



# From subduction to collision in the Parautochthon and autochthon of the NW Variscan Iberian Massif

Francisco J. Rubio Pascual[1], Luis M. Martín Parra[1], Pablo Valverde-Vaquero[1], Alejandro Díez Montes[2], Manuel P. Hacar Rodríguez[3][†], Justo Iglesias[3], Rubén Díez Fernández[4], Gloria Gallastegui[5], Nemesio Heredia[5], L. Roberto Rodríguez Fernández[1]

[1]Instituto Geológico y Minero de España, Madrid, 28003, Spain. e-mail: f.rubio@igme.es
[2]Instituto Geológico y Minero de España, Salamanca, 37001, Spain
[3]Ferrovial Agroman S.A., Madrid, 28046, Spain
[4]Departamento de Geodinámica, Estratigrafía y Paleontología, Universidad Complutense, Madrid, 28040, Spain
[5]Instituto Geológico y Minero de España, Oviedo, 33005, Spain

*Correspondence to*: Francisco J. Rubio Pascual (f.rubio@igme.es)

**Abstract.** A stacking of nappes forms the Parautochthonous Domain located beneath the Allochthonous Complexes of the NW Iberian Massif. Lower Paleozoic metasedimentary and felsic metavolcanic rocks, including some riebeckite gneiss, form the uppermost nappe of the Parautochthon. Micaschists of this nappe may contain albite porphyroblasts with aligned mineral microinclusions defining a relic internal schistosity. Thermobarometric estimates on the relic mineral assemblage suggest an early M1 stage of High-Pressure, Low-Temperature (HP-LT) metamorphism (11-14 kbar; 450-500 ºC), probably related to the same continental subduction process that affected the overlying Lower Allochthon nappes or Basal Units. This uppermost nappe was emplaced onto its relative autochthon by a broad, ductile shear zone that shows top-to-the-E/ENE shear-sense and whose exposure has lateral continuity through to the basal thrust of the Lower Allochthons in the Bragança Complex. The HP-LT metamorphism and its hanging wall position relative to the basal thrust of the Allochthonous Complexes, suggest a re-interpretation of this uppermost parautochthonous nappe as another nappe of the Lower Allochthon. In turn, the other parautochthonous nappes comprise both, Lower Paleozoic pre-orogenic rocks, with no M1 relics, and Upper Paleozoic syn-orogenic rocks. Shear-sense criteria related to the current juxtaposition of the Parautochthon Domain onto the Central Iberian autochthon show contrasting top-to-the-S/SE kinematics. The whole orogenic section was affected by an M2 episode of recrystallization under intermediate P/T conditions during the first stages of collision. P-T estimates on kyanite-bearing schists from the upper sections of the autochthon (9.0 kbar; 425-450 ºC) are consistent with a geothermal gradient somewhat lower than the classical Barrovian one at the beginning of collision. Thermobarometric calculations on schists from the lower parautochthonous nappes yield peak metamorphic conditions around 7.5 Kbar and 600-700 ºC, constraining the original thickness of the allochthonous/parautochthonous pile to about 22.5-27.0 km. Schists sampled from deeper sections of the autochthon yield M2 conditions around 11-12 kbar and 700-725 ºC, matching those registered by correlative regions from Central Iberia. Subsequent M3 syn-collisional recrystallization under High-Temperature and Low-Pressure conditions is associated with top-to-the-N/NW and top-to-the-S/SE extensional flow. The early metamorphic



evolution of the autochthons, Parautochthon and Allochthonous Complexes of NW Iberia recorded a transition in P/T regimes and appears as a model case of a change from subduction to collision.

## 1 Introduction

The parautochthonous domains of the European Variscan Orogen represent paleogeographic and geodynamic intermediate pieces that separate a mainland autochthon, resting below, from an upper set of allochthonous terranes with ophiolites and High-Pressure (HP) rocks representing a suture zone or zones (Matte and Burg, 1981; Arenas et al., 1986; Farias et al., 1987; Martínez Catalán, 1990; Ribeiro et al., 1990; Franke, 2000; Ballèvre et al., 2009; Díez Fernández et al., 2012a; Farias et al., 2014). In contrast to the Early Variscan subduction-related HP-LT metamorphism that affected the overlying Basal Units of the Lower Allochthon in NW Iberia (Munhá et al., 1984; Gil Ibarguchi and Dallmeyer, 1991; Arenas et al., 1995, 1997; Gil Ibarguchi, 1995; Martínez Catalán et al., 1996; Rodríguez et al., 2003; Balcázar et al., 2005; López Carmona et al., 2013) and other correlatable Variscan suture outcrops of the Iberian Massif (Díez Fernández and Arenas, 2015; Arenas et al., 2016a; 2016b; Díez Fernández et al., 2016; Abati et al., 2018), the parautochthonous units are commonly thought not to have been involved in the continental subduction process. Their noticeable imprints of intermediate to low-pressure metamorphism correspond to the subsequent collisional tectonics that also affected the autochthonous domains (e.g. Martínez Catalán et al., 1996; Konopásek et al., 2001; Bellot and Roig, 2007). However, some works (Pitra et al., 2010; Žáčková et al., 2010; Dias Da Silva et al., 2014 discussing Schermerhörn and Kotsch, 1984) and new data from the NW of the Iberian Massif, point to a more complex scenario in which part of the considered parautochthonous section experienced HP-LT conditions before the onset of Barrovian metamorphism. Moreover, some sections of the autochthon immediately beneath the parautochthonous nappes also preserve evidences for an initial geothermal gradient lower than classical Barrovian (Díez Montes, 2007).

Reconstruction of the tectono-thermal structure of the Parautochthon in the NW Iberian Massif is complicated by several reasons. The first of them is the dispersion of isolated outcrops in tectonic klippen between gneiss domes exposing the underlying autochthon in their cores and the massive melting and intrusions of Variscan granitoids. The second reason is the political border between Portugal and Spain (Fig. 1), that historically has compartmentalized the regional research. As a result, even cross-border correlation of first-order tectonic structures has experienced limited advances. We present here the results of new field studies on the tectono-stratigraphic framework of the region and a tectono-metamorphic study of selected rocks from the parautochthonous sections and its nearby autochthons, illustrating the transition from subduction to collision in this part of the European Variscan Belt.



## 2 Geological setting

The NW of the Variscan Iberian Massif is formed by an allochthonous/parautochthonous tectonic pile (Ries and Shackleton, 1971), the so-called Galicia – Trás-os-Montes Zone (GTMZ, Farias et al., 1987; Fig. 1), emplaced onto the autochthonous Central Iberian Zone (CIZ) in Early Carboniferous times (Dallmeyer et al., 1997).

### 2.1 The Galicia – Trás-os-Montes Zone in overview

The GTMZ is divided into the Domain of the Allochthonous Complexes, structurally above, and the parautochthonous Schistose Domain, below.

### 2.1.1 The Allochthonous Complexes

The Allochthonous Complexes comprise three main sets of tectono-metamorphic units (Arenas et al., 1986, 2016b). The top is occupied by an Upper Allochthon of Upper Proterozoic-Lower Paleozoic metasediments and Cambrian intrusives with HP-HT and MP metamorphic recrystallizations, Early Devonian in age. Beneath, it is located a Middle Allochthon with ophiolites of both, Cambrian and Early Devonian protolith ages. At the base it is found a Lower Allochthon, also referred to as the Basal Units, of Early Paleozoic metasediments, metabasites and per-alkaline to calc-alkaline orthogneisses, affected by a Late Devonian HP-LT/MT metamorphic recrystallization (M1, Arenas et al., 1995; Abati et al., 2010), a Lower Carboniferous MP recrystallization (M2) and a Middle Carboniferous HT-LP overprinting (M3). The current allochthonous pile was built during Devonian and Early Carboniferous times by the eastwards stacking of previously subducted and detached slabs. One of the main structures of this stage is the Lalín-Forcarei Thrust (Martínez Catalán et al., 1996; Fig.2), which continues up to the basal thrust of the Centro-Trasmontan Domain (Ribeiro, 1974), through the Fumaces Thrust (in the Verín Synform area, this work), emplacing the allochthonous pile onto the Parautochthonous Domain (Fig. 3).

### 2.1.2 The Parautochthon

The parautochthonous Schistose Domain in Spain (Farias et al., 1987) or Peri-Trasmontan Domain in Portugal (Ribeiro, 1974) is mainly formed by Palaeozoic metasediments and some metavolcanic rocks (Díez Fernández et al., 2012a). In an overall approach, the rocks have undergone the Barrovian MP metamorphism (M2) reaching medium-grade conditions in the more internal sections of the GTMZ pile. HT-LP overprinting (M3) has also been more intense in structurally deep and internal positions. Previous interpretations of a continuous stratigraphic succession (Marquínez, 1984; Farias, 1990; Toyos, 2003) have changed to recent models of two imbricated tectono-stratigraphic groups of rocks. Both units are referred to as the Lower Parautochthon, made of pre-orogenic and syn-orogenic materials forming thrust-sheets, and the Upper Parautochthon with mostly pre-orogenic rocks forming fold-nappes (Rodrigues et al., 2003, 2006). Recent works have also pointed out the importance of Late Palaeozoic syn-orogenic rocks forming part of the Lower Parautochthon (Martínez Catalán et al., 2008, 2016; Meireles, 2011; Dias da Silva et al., 2014a, 2014b; González Clavijo et al., 2016).





The complete allochthonous/parautochthonous pile was piggyback transported to its current position above the CIZ autochthons by a new generation of thrusts. One of the main structures (Figs. 1, 3) is the Verín Thrust (Farias et al., 1987; Farias, 1990) and its prolongation in the basal thrusts of the Peri-Trasmontan Domain (sometimes recognized as the Main Trás-os-Montes Thrust), and the Alcañices Synform. The Verín Thrust has been dated at 340.0 ± 0.9 Ma and the main thrust in Alcañices at 342.6 ± 0.3 Ma (whole-rock $^{40}$Ar/$^{39}$Ar analyses, Dallmeyer et al., 1997).

The thickened crustal wedge was then thermally weakened and partially extended (Alcock et al., 2009; Martínez Catalán et al., 2009; 2014; Díez Fernández et al., 2012b) around the middle of the Carboniferous times (extensional stage E1, Alcock et al., 2009; 2015). Some of the main structures are the top-to-the-NW Bembibre-Pico Sacro Detachment (Martínez Catalán et al., 2002; Gómez Barreiro et al., 2010), the top-to-the-SE Redondela-Bearíz Detachment (Díez Fernández et al., 2012b) and the top-to-the-NW Arnoia Detachment (this work, Fig. 2). Detachments are related to extensional flow, and the development of gneiss domes with the autochthonous CIZ outcropping in their cores (Padrón and Bande-Celanova domes, Fig. 1, and probably also the Tabagón Anticline in the Monteferro area) seems to be nucleated between pairs of conjugate structures (Díez Fernández, 2011). The regional pattern of Figure 1 also shows the allochthonous klippen preserved in upright C3 synforms and the autochthons cropping out in tectonic windows in the core of the C3 antiforms.

## 2.2 The autochthonous Central Iberian Zone

The mostly pre-orogenic rocks of the CIZ crop out structurally below, inside tectonic windows or around the GTMZ. They are formed by Upper Proterozoic to Lower Devonian metasediments and Upper Cambrian to Lower Ordovician metavolcanic and metagranitic orthogneisses. These last ones are related to the calc-alkaline peraluminous magmatism of the Ollo de Sapo Formation (Lancelot et al., 1985; Bea et al. 2006; Díez Montes, 2007; Montero et al., 2009a; Díez Montes et al., 2010; Talavera et al., 2013). The rocks are affected by E-vergent recumbent folds (C1) with an associated axial-planar schistosity S1. The GTMZ is usually thrusting (C2) onto the already folded Upper Ordovician to Silurian strata of the CIZ, crosscutting the C1 folding structure (Farias, 1987, 1990; Marcos and Farias, 1999; Díez Montes, 2007), or it locally thrusts onto syn-orogenic deposits covering the CIZ autochthon in the Alcañices Synform (González Clavijo, 2006; González Clavijo and Martínez Catalán, 2002; González Clavijo et al., 2016; Farias and Marcos, 2019; Fig. 3). In deep shear zones of the CIZ, such as in the core of gneiss domes and C3 antiforms, S1 is also transposed by a flat-lying foliation S2. Stretching lineations and asymmetric microfolds in this foliation indicate a top-to-the-SE shear-sense during C2 ($L_{C2}$) and probably E1, while upper crustal sections (here represented by the GTMZ) show top-to-the-NW shear-sense ($L_{E1}$). Similar E1 extensional shear zones are widespread represented along the most internal zones of the CIZ (Doblas et al., 1994; Escuder Viruete et al., 1994, 1998; Díez Balda et al., 1995; Arango et al., 2013; Díez Fernández et al., 2013; Rubio Pascual et al., 2013, 2016; Díez Fernández and Pereira, 2016). From the metamorphic geology point of view, the materials of the CIZ cropping out from immediately around the GTMZ to the central areas of the Iberian Massif are characterized by the onset of thick and complete Barrovian sequences, which are variably overprinted by the HT-LP metamorphism (Martínez Catalán et al., 2014).



## 3 Tectonostratigraphy of the parautochthonous Schistose Domain

The parautochthons in NW Spain have been traditionally subdivided into three pre-Variscan stratigraphic groups, from
bottom to top: Santabaia (Farias et al., 1987), Nogueira and Paraño (Marquínez, 1981, 1984). They were firstly defined in
the Schistose Area of Central Galicia (SACG, Fig. 1) and in the A Seara Synform area, and then, some or all of them,
extended to other isolated outcropping areas of the NW Iberian Massif (Verín Synform, Marquínez, 1984; Farias et al., 1987;
Chantada area, Barrera Morate et al., 1989; Marcos and Llana Fúnez, 2002). Other parautochthonous sequences have been
defined in other areas, such as beneath the Cabo Ortegal Complex (Marcos and Farias, 1999; Marcos et al., 2002), and in the
Monteferro-El Rosal area (Toyos, 2003; Llana-Fúnez, 2001). The presence of Silurian graptolite fauna in rocks from the
Nogueira Group (or similar graphite-rich black chert and schists formations, see review by Piçarra et al., 2006) for long time
has sustained a supposedly continuous Ordovician to Devonian age for the whole parautochthonous sequence. However, the
U-Pb ages that have been more recently obtained in volcanic rocks support an at least Early Ordovician to Silurian age for
the structurally upper group or groups of rocks (Valverde-Vaquero et al., 2005, 2007), probably due to some thrusting of
Ordovician rocks (e.g. in Cabo Ortegal, Rodríguez et al., 2004; Valverde-Vaquero et al., 2007).

The most likely interpretation obviously suggests that the Parautochthon in NW Spain must have tectonostratigraphic
characteristics similar to those described in Portugal. In the Portuguese region of Trás-os-Montes, the parautochthonous
series of the Peri-Trasmontan Domain (Ribeiro, 1974) have been early described as a thick tectonic imbricate of
Ordovician/Silurian to Devonian formations (also named Parautochthonous Thrust Complex, Ribeiro et al., 1990). The
imbricate is placed between two main thrusts: the Main Trás-os-Montes Thrust (Ribeiro et al., 1990, the basal
parautochthons thrust) and the Centro-Trasmontan Thrust (the basal allochthons thrust) at its top. It was later subdivided into
a Lower and an Upper Parautochthon with different tectonic and stratigraphic characteristics (Meireles et al., 1995;
Rodrigues et al., 2003). The Lower Parautochthon could include Ordovician to Silurian pre-orogenic formations, but Upper
Devonian to Early Carboniferous syn-orogenic rocks mainly form it (Pereira et al., 1999; Meireles, 2011; Dias da Silva et al.,
2014b). The Upper Parautochthon includes Upper Cambrian-Late Ordovician calc-alkaline metavolcanic rocks (Dias da
Silva et al., 2014a, 2016) and Ordovician to Silurian metasediments.

Some recent works have proposed that the Main-Trás-os-Montes Thrust is actually located between the Upper and the Lower
Parautochthon. The real GTMZ basal fault would be a different one, named Basal Lower Parautochthonous Detachment,
below the Morais Allochthonous Complex (Dias da Silva et al., 2014a, b, 2016), correlatable with the Basal Thrust of the
parautochthons in the Alcañices Synform (González Clavijo and Martínez Catalán, 2002). This interpretation considers most
of the former HP-LT Lower Allochthon as a part of the Upper Parautochthon, thus introducing some tectonometamorphic
problems. First, because there is not any other recognized basal thrust structure for the Lower Allochthon of the Bragança
Complex. Second, because some of the HP-LT rocks of the Morais Complex stay in the Upper Parautochthon. However,
there is no doubt about the correlation of the basal shear zone of the Parautochthonous Thrust Complex with the Verín
Thrust (Farias, 1990).





Anyway, a major problem of cross-border correlation existed up to date, as the Centro-Trasmontan Thrust (no matter whether it is at the base of the Bragança Allochthonous Complex or between the Lower and Upper Parautochthon) was not previously recognized in Spain. For drawing purposes of small-scale regional schemes, it was sometimes used the simplification of an appropriate normal lithostratigraphic contact taken from Alonso Alonso et al. (1981) and Nuño Ortea

(1981). According to our fieldwork, the uppermost part of the Paraño Group is tectonically placed onto the rest of the same group through the Fumaces Thrust, being a continuation of the Centro-Trasmontan Thrust. The Fumaces Thrust is formed by a near two hundred meters-thick shear zone made of phyllonitic schists, quartzites and variably deformed bodies of alkaline metavolcanics, ranging from preserved trachytic textures to L-S mylonites. Quartz veinlets, K-feldspar porphyroclast tails and riebeckite acicular prisms orientations mark an E-W to ENE-WSW stretching lineation $L_{C1}$ (Fig. 4a). Kinematic criteria

for the fabrics formed during thrusting are orthogonal to the Variscan trend and consistent with both, the tectonic stacking of the allochthonous Basal Units (Lalín – Forcarei Thrust), and the first contractive deformations in the autochthons (C1, Alcock et al., 2009). For this reason, we will refer to their foliation as $S_{C1}$, independently of whether it is not the first schistosity developed in these rocks. As we will further see, this uppermost nappe of the Verín Synform and other upper parts of the currently considered parautochthonous section, such as the Paraño Group in the SACG, present igneous, petro-

tectonic and metamorphic characteristics different to those of the underlying part of the section.

On the other hand, the Verín Thrust also presents a thick low-angle shear zone affecting the base of the Parautochthonous Domain (Farias, 1990), but with systematic top-to-the-SE/ESE kinematic criteria, such as asymmetric boudins and drag folds, stretching mineral lineations and stretching elongated pebbles ($L_{C2}$). Right to the north, the correlatable shear zone below the Cabo Ortegal Complex, (Rio Baio Thrust Sheet, Marcos et al., 2002) shows consistent $L_{C2}$ top-to-the-S criteria at

its basal levels (Figs. 1, 5), following the arcuate shape of the regional structures. These structures are kinematically correlatable to the "out-of-sequence thrusts" (Martínez Catalán et al., 2002; D4 in Gómez Barreiro et al., 2006) of the allochthonous complexes. It is important to notice that the overthrusting direction of this second contractional episode (C2) is contrastingly near parallel to the Variscan trend, so the change from C1 to C2 is not a simple progradation of the tangential deformation. The upper part of the parautochthonous section below the Cabo Ortegal Complex (e.g. Arenas et al., 2009), has

been also considered an allochthonous Basal Unit (Ramallal Phyllonites, Marcos and Farias, 1999, Figs. 1, 5). The rocks in this high-strain zone show mineral stretching lineations with top-to-the-NNE shear-sense that we put into relation to E1 ($L_{E1}$). According to our fieldwork, renewal of orthogonal contractive deformation developed a set of discrete (1-3 m-thick) shear zones related to higher angle (45º) top-to-the-ENE thrusts that are crosscutting even the $S_{C2}$ schistosity. Marcos and Farias (1999) describe several kinematic indicators of NE-directed thrusting in Cabo Ortegal. These thrusts seem to have

reactivated the contact zone between Parautochthon and autochthon and they present a similar pattern to the top-to-the-NE stacking imbricate beneath the Alcañices Synform (González Clavijo, 2006), that is also described as late out-of-sequence thrusts by Farias and Marcos (2019).

The mechanical contact between the allochthonous Basal Units and the underlying Parautochthon or autochthon in the most internal zones of the GTMZ is affected by extensional detachments (E1) crosscutting the stacking structure (Martínez



Catalán et al., 2002). The same detachment system affects the contact between the Parautochthon and the underlying autochthon in the Padrón Dome (Díez Fernández et al., 2012b, 2017, 2019) and the Bande-Celanova Dome - A Seara Synform. Lower Ordovician metagranitic orthogneises of the CIZ (Talavera et al., 2013) occupy the core of these migmatitic domes (Martínez Catalán et al., 2014). A succession of Santabaia, Nogueira and lower parts of Paraño groups forms the parautochthonous section. The basal Arnoia Detachment is probably masking the former thrust structure and removed most

of the autochthonous Paleozoic series formerly overlying the orthogneisses (Fig. 2). This low-angle detachment is a >3 km-thick extensional shear zone with top-to-the-N/NW L-S mylonites ($L_{E1}$-$S_{E1}$) at the base and S-C structures to the top.  The band of S-C structures crops out even around all the A Seara Basin. Its petro-tectonic and metamorphic characteristics are treated below.

## 4 Tectonometamorphic constraints.

### 4.1 Barrovian metamorphism.

The whole section of the parautochthon in the SACG and A Seara presents an M2 Barrovian sequence of metamorphic zones (first studied by Minnigh, 1975, and Marquínez and Klein, 1982) which is related to the main episodes of collisional thickening (Martínez Catalán et al., 1996). This zonation is approximately parallel to the C2 regional structure and develops progressively syn-C2 Chl, Bt, Grt, St and Ky (the last one not previously found; mineral abbreviations after Whitney and

Evans, 2010) with increasing structural depth. A narrow M2 inverted metamorphic sequence also developed on the upper structural part of the pile in the SACG, induced by downwards transferred heat from the Upper Allochthons (Martínez Catalán et al., 1996). Mineral growth in the inverted metamorphic zonation even postdates C2, and the post- $S_{C2}$ biotite, garnet, staurolite and sillimanite isograds crosscut the C2 regional structure, but they are affected by the C3 folding, so they were probably developed at the beginning of E1. Medium-pressure conditions were also achieved in the lower structural

levels of the Parautochthon in other central areas of the GTMZ, such as Chantada (Marcos and Llana-Fúnez, 2002) and the Riás schists (Díez Fernández, 2011; Solís-Alulima et al., 2019). In the eastern margin of the GTMZ, from the Cabo Ortegal Complex to the Alcañices Synform, metamorphic M2 conditions are of low to very low grade (Dias da Silva and González Clavijo, 2010). The parautochthonous pile recrystallized under chlorite to biotite conditions in the Verín Synform. Garnet-staurolite-kyanite M2 assemblages are restricted west of the synform. Nonetheless, in the westernmost area of Monteferro-El

Rosal, M2 medium-pressure conditions (Grt + St assemblages in metapelites) affected only the autochthonous Precambrian to Lower Ordovician rocks, while the whole autochthonous-parautochthonous section experienced an important HT-LP overprinting (Toyos and González Menéndez, 2010), probably the extensionally-driven M3.

### 4.2 Pre-Barrovian metamorphism.

A foliation earlier than collisional $S_{C1}$ is preserved as microinclusion trails in albite porphyroblasts from the schists of the

Paraño Group in the SACG (Marquínez, 1984). As we will further see, the internal schistosity of the albite porphyroblasts



include mineral relics of an initial episode of HP-LT metamorphic recrystallization M1, probably correlatable to that one of the immediately overlying Basal Units of the allochthons. Albite-bearing schists are characteristic rocks in the allochthonous Basal Units, where the porphyroblasts present a more advanced textural development and include more complete HP-LT mineral parageneses (Arenas et al., 1995). This HP metamorphism of the Basal Units is related to the process of continental

subduction prior to the setting of collisional tectonics (Arenas et al., 1995; Martínez Catalán et al., 1996).

For several reasons, the HP nappe of the SACG could have a likely continuation in the uppermost part of the Paraño Group in the Verín Synform. Farias (1990) also mentions there the presence of albite porphyroblasts with an internal schistosity of aligned micro-inclusions of quartz, chlorite and white mica. The Fumaces Thrust of Verín has outcropping continuity into the basal thrust of the lower allochthonous Centro-Trasmontan Complex with HP-LT rocks (Ribeiro, 1976; Munhá et al.,

1984; Schermerhörn and Kotsch, 1984; Balcázar et al., 2005). In the same way, alkaline orthogneisses are also present in the hanging-wall of both sections. The rifting-related Ordovician alkaline/per-alkaline magmatism is a characteristic feature of former external zones of the continental margin of Gondwana, later forming the Basal Units (Díez Fernández et al., 2012c), and other possibly correlatable HP-LT units of the Variscan Massif (Díez Fernández et al., 2015).

### 4.3 Low-pressure metamorphism.

Another low-angle foliation $S_{E1}$ is present in the lowest structural levels of the parautochthonous pile around the high-grade gneiss domes where the migmatized rocks of the autochthon crops out, such as Bande-Celanova and Padrón domes, or in the base of the SACG, forming some hundreds of meters-thick extensional shear bands that overprint the former $S_{C2}$. In the Arnoia Detachment (Fig. 2), described above, the M2 Barrovian zonation of the lower structural levels of the parautochthonous pile (Santabaia and Nogueira groups) presents telescoped isograds and narrowed zones by the effect of the

extensional deformation E1, and the rocks were affected by the HT-LP metamorphic overprinting. Sillimanite, andalusite and new biotite grew in the $S_{E1}$ foliation of the lower structural levels of the Arnoia Detachment, while S-C structures and growth of new chlorite affected the upper structural levels of the shear zone developed on the rocks of Paraño sequences. Growth of syn-kinematic andalusite porphyroblasts in metapelites of the Parautochthon has been also described in Chantada (Marcos and Llana-Fúnez, 2002), Monteferro-El Rosal (Toyos and González Menéndez, 2010), below the Morais Complex

(Dias da Silva and González Clavijo, 2010) and the Riás schists (Díez Fernández, 2011; Díez Fernández et al., 2019; Solís-Alulima et al., 2019).

### 5 New metamorphic data and thermobarometric estimations

In order to better understand the metamorphic conditions and tectonothermal evolution that affected the parautochthonus units, we have studied several samples of semipelitic albite porphyroblast-bearing schists from its uppermost section and two

samples of pelitic schist from the lowermost structural levels. We also studied two samples from the nearby autochthons with the objective of constraining the tectonothermal regime and the effective thickening of the collisional wedge: one sample of Ky-bearing, low-grade pelitic schist from the upper structural levels, and one garnet-sillimanite schist from the




mesozonal structural levels below. Characteristics of the samples, thermobarometric methods and P-T results are summarized in Table 1.

## 5.1 Parautochthonous units

As previously outlined in the introduction, it has been commonly accepted that the parautochthonous rocks of the GTMZ had firstly experienced an M2 episode of regional recrystallization under intermediate P/T conditions, typical of collisional stages in orogeny. Contrarily to the overlying Basal Units, it has been supposed that they were not involved in the previous stages of continental subduction and were not thus affected by the M1 episode of HP-LT metamorphism (Martínez Catalán et al., 1996).

However, the uppermost parautochthonous nappe in the SACG and the Verín Synform is mostly formed by pelitic and semipelitic schists, which may contain albite porphyroblasts (Fig. 4b; samples 1 and 2 in Table 2) containing inclusions trails made of Qz, Chl, Tur, Rt, Ilm and rare white mica (WM). These minerals define an internal schistosity ($S_i$) that is previous to the main external schistosity ($S_e = S_{C1}$). WM from the $S_i$ show high silica contents (Table 2) and a mineral chemistry compatible with HP-LT metamorphic conditions. Combined Massonne and Schreyer (1987) Si-in-phengite geobarometer and Pl-Ms geothermometer (Green and Usdansky, 1986) suggest minimum pressure conditions around 11-14 kbar and temperatures of 450-500 ºC for the early metamorphic recrystallization of these rocks (Fig. 6). Furthermore, the alkaline metavolcanic rocks of this uppermost nappe present metamorphic growth of acicular riebeckite around relic (igneous?) aegirine crystals. These observations point to an early burial under a HP gradient and a subsequent syn-collisional exhumation during C1.

There are no albite porphyroblast-bearing schists in the lower structural levels of the Parautochthon. In turn, the Barrovian episode of metamorphic recrystallization M2, and the HT-LP M3, affected those rocks with variable intensity. The higher conditions seem to be achieved in central areas of the GTMZ pile. We have carried out THERMOCALC v.3.33 (Powell and Holland, 1988; Holland and Powell, 2011) average P-T estimations for the assemblage Grt + Bt + Pl + Ms + $Al_2SiO_5$ + Rt + Ilm on two samples of pelitic schists from the Lower Parautochthonous nappe (samples 3 and 4, mineral chemistry data in Table 3). Sample 3 was collected in the lower section affected by the extensional Arnoia Detachment (Fig. 4c), and yield an isobaric heating from 595 ºC ± 22 at 7.4 ± 0.9 Kbar to 696 ± 25 ºC at 7.6 ± 0.6 Kbar (fields 3c-Grt center, 3r-Grt rim, respectively in Fig. 6). Sample 4 was taken 100 m above the previous one, and the P-T values of the rock show a small retrograde path from 663±26 ºC at 6.6 ± 1.1 Kbar to 637 ± 27 ºC at 5.4 ± 1.1 Kbar (fields 4c-Grt center, 4r-Grt rim, respectively, in Fig. 6) which is consistent with the final cooling (M3) in the extensional detachment.

## 5.2 Autochthons

In the autochthons of the Sanabria region, some low-grade metapelites of the structural upper sections (close beneath the GTMZ tectonic pile) are formed by Ky-bearing (Late Ordovician) schists and veins that show low-temperature assemblages with high-silica WM, Chl, Rt, Il and Ap (Fig. 4d, sample 5 in Table 2). The rocks are Bt and Pl-free. The Si-in-Phe





geobarometer (Massonne and Schreyer, 1987) yields minimum pressures of 8.5 kbar for temperatures that cannot be much higher than the pyrophillite = Ky + Qz + $H_2O$ reaction (Winkler, 1976), proceeding at 425-450 ºC (Fig. 6). On the other hand, the Early Ordovician Grt-Sill schists from the Villadepera Antiform (sample 6) are representative of mid-crustal sections of the autochthons with Barrovian metamorphism. THERMOCALC average P-T analyses on the mineral assemblage displaying Grt + Bt + Pl + Ms + $Al_2SiO_5$ + Rt + Ilm (Fig. 4e, mineral chemistry data in Table 4), give conditions

near 11-12 kbar and 700-725 ºC (6c-Grt center, 6r-Grt rim, Fig. 6).

## 6 Discussion and conclusions

Some parts of the Upper Parautochthon have undergone a HP-LT episode of metamorphic recrystallization (M1: 450-500 ºC; 11-14 kbar; geothermal gradient about 13 ºC/km) compatible with a continental subduction process. The HP-LT rocks in the parautochthons are restricted to the uppermost nappe in the SACG and its probable continuation in the Verín Synform,

where alkaline riebeckite-bearing gneisses are present. It implies the existence of a major contractive structure localized at the base of this nappe that we have identified in the Fumaces Thrust, in continuation of the basal thrust of the Bragança Allochthonous Complex (Trás-os-Montes). The previously subducted rocks were eastwards exhumed and emplaced onto the other regions of the current Upper Parautochthon (C1). Alternatively, in accordance with Ribeiro (1974), this uppermost parautochthonous nappe could be considered the lowest of the (allochthonous) Basal Units in the region.

Another nappe with Early Ordovician alkaline metavulcanites occupies the uppermost structural position of the Parautochthons in Cabo Ortegal Complex, possibly thrusting on Middle Ordovician slates (Valverde-Vaquero et al., 2005). Those Middle Ordovician slates and the rest of the Paraño Group (lower sections without M1 recrystallization evidences, including some felsic metavolcanic rocks, Furongian-Lower Ordovician in age; Valverde-Vaquero, pers. comm.) are also thrusting above the Nogueira Group, whose rocks present Silurian fauna in Cabo Ortegal, Verín and other areas (Piçarra et

al., 2006). Therefore, another consequence is that the rest of the Upper Parautochthon is formed by one or more nappes made of Upper Cambrian to Ordovician rocks, and possibly Silurian to Devonian rocks (e.g. Ribeiro, 1974; Meireles, 2011).

In the Lower Parautochthon, lydites and graphitic schists with Silurian faunal sites (Nogueira Gp) form large fold-nappes, sometimes with Lower Ordovician orthogneises (Santabaia type) in their core (Cabo Ortegal area, A Seara Synform). Syn-orogenic sediments were deposited unconformably onto the Lower Parautochthon and at the front of the nappes (Martínez

Catalán et al., 2016). In the Verin Synform they include fragments of lydites and other rocks from their immediate substrate, but also fragments of alkaline trachytes from the uppermost nappe (Fig. 4f). They were progressively involved in the thrust-and-nappe tectonics (Martínez Catalán et al., 2016), showing invariably along-trend top-to-the-S/SE kinematics (C2) related to the emplacement of the GTMZ on the CIZ autochthons. The series of syn-orogenic sediments in the Spanish side (Verín) thicken towards more external zones (E and SE directions), becoming the main part of the materials column in the Lower

Parautochthon of Trás-os-Montes (Rodrigues et al., 2003) and Alcañices (González Clavijo et al., 2016). Thus, differences in the style of deformation between Lower Parautochthon of thrust-sheets and Upper Parautochthon of fold-nappes, proposed



by Rodrigues et al. (2003) for the Trás-os-Montes region, seem to be related to differences in deformational history and/or rheology (syn-orogenic vs pre-orogenic materials).

P-T calculations on pelitic rocks with Barrovian assemblages affected by E1 extensional detachments yield conditions of 7.5 Kbar and 600-700 ºC (geothermal gradient about 29 ºC/km) for the Lower Parautochthon in central parts of the GTMZ. It implies thickness above 22.5 km for the whole GTMZ pile at some initial moment of syn-collisional extension (change from M2 to M3). However, conditions of around 9.0 kbar and 425-450 ºC (geothermal gradient of 16 ºC/km), calculated for Ky-bearing rocks of the low-grade, structurally upper levels of the autochthons, possibly indicate up to near 27 km thickness of the GTMZ at the beginning of collisional tectonics. The P-T values calculated on nearby rocks of the autochthon (11-12 kbar

and 700-725 ºC; geothermal gradient about 21 ºC/km) match those registered by correlative rocks from Central Iberia. The P-T trajectories of medium and high-grade rocks from deep sections of the NW Iberian Massif (Lower Parautochthon in the Arnoia Detachment, CIZ in the Villadepera Antiform, respectively) are quite similar to those from Central Iberia (paths CI-1 and CI-2, respectively, in Fig. 6; Rubio Pascual et al., 2013).

The geothermal gradient inferred from the low-grade rocks from the autochthon at the top of the footwall (early M2: 16

ºC/km) is higher than the subduction-related gradient (M1: 13 ºC/km), but somewhat lower than the classical Barrovian one of the deeper section of the autochthon (M2: 21 ºC/km; late M2 to early M3: 29 ºC/km). This metamorphic record is possibly a model case of transition from HP/LT regimes in subduction systems to medium P/T regimes in collisional situations.

*Financial support*. This research has been supported by the Spanish Ministerio de Economía, Industria y Competitividad

(grant nos. CGL2016-76438-P and IGME 2281).

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







**Figure 1.** Tectonic scheme of the NW Iberian Massif, with indication of sampling locations for thermobarometry and alkaline/per-alkaline rocks localities. A, Alcañices Synform; AS, A Seara Synform; B, Bragança Complex; BC, Bande-Celanova Dome; Ch, Chantada area; CO, Cabo Ortegal Complex; M, Morais Complex; MER, Monteferro – El Rosal area; O, Ordenes Complex; P, Padrón Dome; R, Riás schists; S, Sanabria; SACG, Schistose Area of Central Galicia; V, Verín Synform; Vi, Villadepera Antiform.



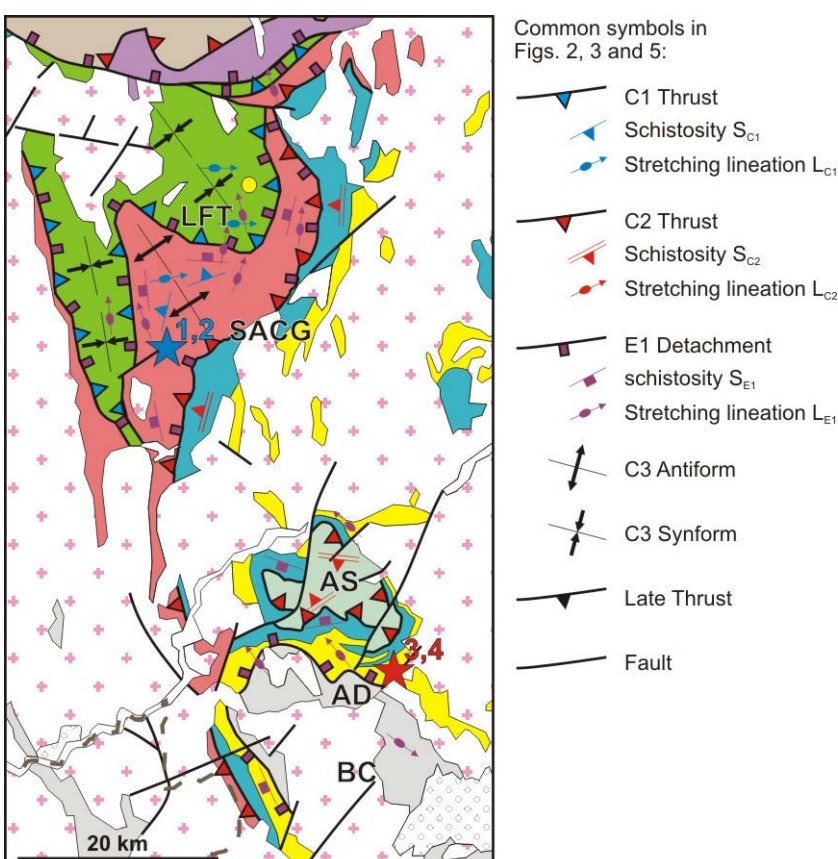

Common symbols in
Figs. 2, 3 and 5:

— ▽ C1 Thrust
◀ Schistosity $S_{C1}$
•➤ Stretching lineation $L_{C1}$

— ▼ C2 Thrust
◀ Schistosity $S_{C2}$
•➤ Stretching lineation $L_{C2}$

— ■ E1 Detachment
■ schistosity $S_{E1}$
•➤ Stretching lineation $L_{E1}$

↕ C3 Antiform

⤹ C3 Synform

— ▼ Late Thrust

— Fault

**Figure 2.** Tectonic scheme of the Parautochthon in the Schistose Area of Central Galicia. The stretching lineation arrows indicate direction of top displacement. AD, Arnoia Detachment; LFT, Lalín-Forcarei Thrust. Rest of 630   the legend as in Fig. 1.



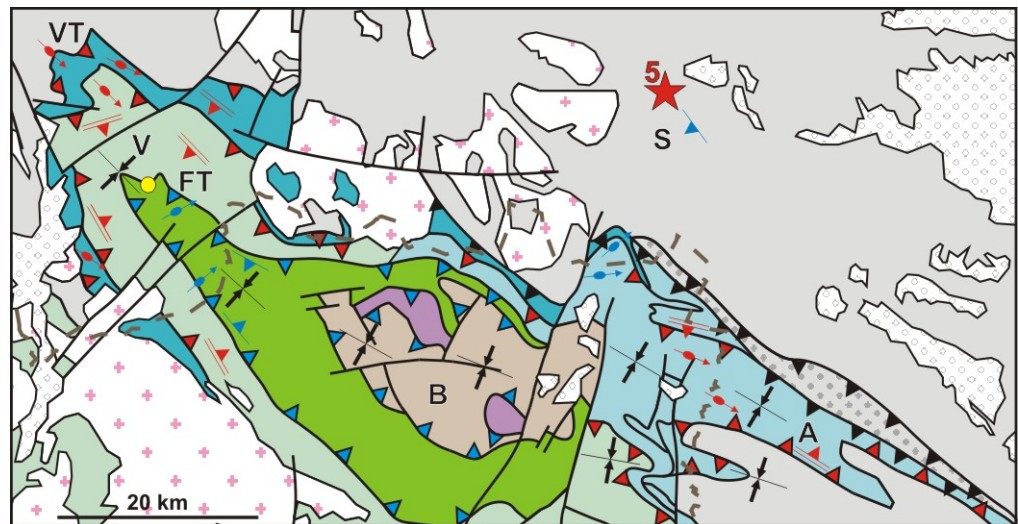

**Figure 3.** Tectonic scheme of the Parautochthon in the Verín Synform – Bragança Complex – Alcañices Synform area. FT, Fumaces Thrust, VT, Verín Thrust. Rest of symbols as in Fig. 1 and Fig. 2.






**Figure 4.** Petrographical aspects of the rocks. **a)** Peralkaline gneiss, Verin Synform. K-feldspar porphyroclasts in a matrix of feldspar, quartz and oriented crystals of blue riebeckite. Other samples can show radial growth of riebeckite around dark green small crystals of aegirine. **b)** Albite porphyroblast-bearing schist, SACG. Microinclusions in the internal schistosity can be formed by quartz, white mica, chlorite, rutile, ilmenite, tourmaline and organic matter. **c)** Garnet-kyanite-sillimanite schist, Viladepera Antiform. **d)** Ky-bearing schist, Sanabria. Most of this rock is formed by kyanite, white mica and rutile. **e)** Garnet-sillimanite schist, lower section of the Arnoia Detachment. **f)** Syn-orogenic conglomeratic sandstone, Verin Synform. Fragments are formed by quartz, quartzite (Qtzt), slate, schist (Sch) and trachyte (Trch).





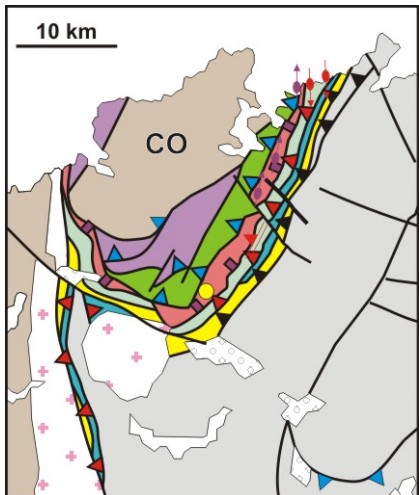

**Figure 5.** Tectonic scheme of the Parautochthon in the Cabo Ortegal area. Same legend as Fig. 1 and Fig. 2.



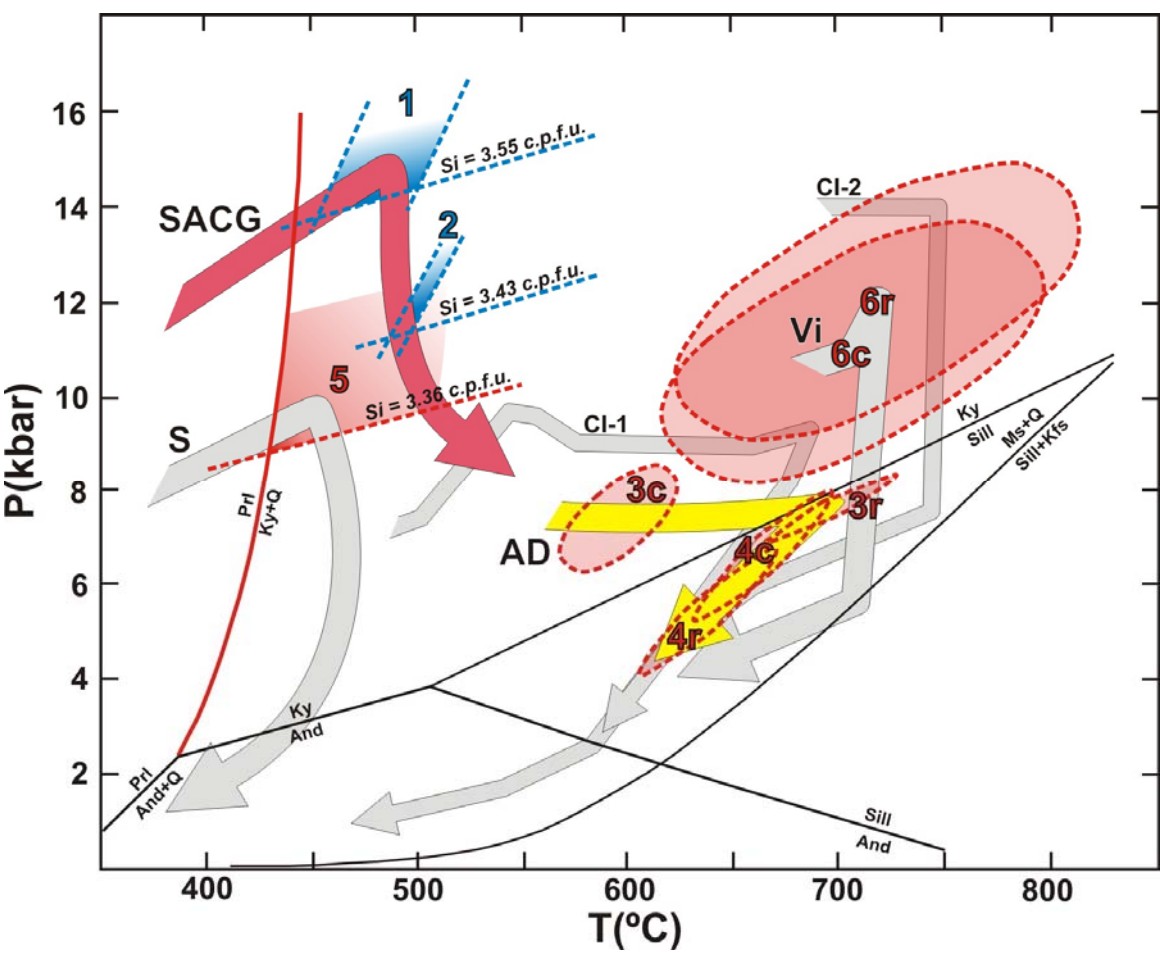

**Figure 6.** Thermobarometric results and inferred P-T paths for the studied samples. Same legend as in Fig. 1 and Fig. 2.
CI-1 and CI-2 paths of autochthonous sections from Central Iberia (after Rubio Pascual et al., 2013) are shown for comparison purposes.





| Unit | Rock sample | Met. event | Mineral assemblage | Barometer | Thermometer | P (kbar) | T (ºC) |
|---|---|---|---|---|---|---|---|
| Upper Parautochthon or Lower Allochthon | (1) Ab-schist | M1 | Ab+Ph+Chl+Rt+Il+Qz | Si-in-Ph (MS87) | Pl-Ms (GU86) | >14.1±0.3 | 477±24 |
| | (2) Ab-schist | M1 | Ab+Ph+Chl+Rt+Il+Qz | Si-in-Ph (MS87) | Pl-Ms (GU86) | >11.3±0.1 | 493±50 |
| Lower Parautochthon | (3c) Grt-Sil schist | M2 | Grt-c+Ms+Bt+Pl+AlS+Rt+Il+Qz | THERMOCALC v.3.33 (HP98,HP11) | | 7.4±0.9 | 595±22 |
| | (3r) Grt-Sil schist | M2 | Grt-r+Ms+Bt+Pl+Sil+Il+Qz | THERMOCALC v.3.33 (HP98,HP11) | | 7.6±0.6 | 696±25 |
| | (4c) Grt-Sil schist | M3 | Grt-c+Ms+Bt+Pl+Sil+Il+Qz | THERMOCALC v.3.33 (HP98,HP11) | | 6.6±1.1 | 663±26 |
| | (4r) Grt-Sil schist | M3 | Grt-r+Ms+Bt+Pl+Sil+Il+Qz | THERMOCALC v.3.33 (HP98,HP11) | | 5.4±1.1 | 637±27 |
| Central Iberian Zone (Upper Ordovician) | (5) Ky-slate | M2 | Ky+Ph+Chl+Rt+Ap+Qz | Si-in-Ph (MS87) | Prl-out reaction (W76) | >8.5 | >450 |
| Central Iberian Zone (Lower Ordovician) | (6c) Grt-Sil schist | M2 | Grt-c+Ms+Bt+Pl+AlS+Rt+Il+Qz | THERMOCALC v.3.33 (HP98,HP11) | | 10.9±2.2 | 705±73 |
| | (6r) Grt-Sil schist | M2 | Grt-r+Ms+Bt+Pl+AlS+Rt+Il+Qz | THERMOCALC v.3.33 (HP98,HP11) | | 12.0±2.3 | 720±77 |

**Table 1.** Resume of mineral assemblages, thermobarometric methods and results. GU86, Green and Usdansky (1986); HP98, Holland and Powell (1998); HP11, Holland and Powell (2011); MS87, Massonne and Schreyer (1987); W76, Winkler (1976).





| Unit | Uppermost Parautochthon/Lowermost Allochthon | | | | | | Upper section of the CIZ |
|---|---|---|---|---|---|---|---|
| Location | SACG | | | | | | Sanabria |
| Sample | 1 (Ab-porphyroblast schist ) | | | 2 (Ab-porphyroblast schist ) | | | 5 (Ky-bearing schist) |
| Analysis | Ms44 | Pl42 | Pl55 | Ms56 | Pl51 | Pl59 | Ms255 |
| $SiO_2$ | 53.29 | 68.69 | 68.19 | 51.82 | 64.85 | 64.43 | 52.44 |
| $TiO_2$ | 0.14 | 0.05 | 1.15 | 0.28 | 0.00 | 0.00 | 0.32 |
| $Al_2O_3$ | 28.64 | 19.66 | 19.10 | 30.43 | 20.61 | 21.48 | 33.32 |
| $Cr_2O_3$ | 0.00 | 0.00 | 0.01 | 0.00 | 0.00 | 0.00 | 0.00 |
| FeO | 0.72 | 0.00 | 0.01 | 0.85 | 0.09 | 0.18 | 1.04 |
| MnO | 0.01 | 0.01 | 0.00 | 0.01 | 0.03 | 0.02 | 0.03 |
| NiO | 0.05 | 0.01 | 0.00 | 0.00 | 0.00 | 0.00 | 0.00 |
| MgO | 0.36 | 0.00 | 0.02 | 0.30 | 0.02 | 0.00 | 0.58 |
| CaO | 0.00 | 0.29 | 0.04 | 0.00 | 1.93 | 2.32 | 0.27 |
| $Na_2O$ | 0.92 | 11.50 | 12.06 | 0.84 | 10.26 | 10.96 | 1.92 |
| $K_2O$ | 7.79 | 0.04 | 0.06 | 8.47 | 0.06 | 0.06 | 5.32 |
| $P_2O_5$ | 0.01 | 0.01 | 0.03 | 0.03 | 0.02 | 0.00 | 0.01 |
| F | 0.07 | 0.01 | 0.00 | 0.00 | 0.01 | 0.04 | 0.08 |
| Cl | 0.00 | 0.01 | 0.00 | 0.00 | 0.00 | 0.01 | 0.01 |
| Total | 91.97 | 100.27 | 100.67 | 93.05 | 97.86 | 99.47 | 95.29 |
| | O = 11 | O = 8 | O = 8 | O = 11 | O = 8 | O = 8 | O = 11 |
| Si | 3.55 | 2.99 | 2.97 | 3.43 | 2.93 | 2.88 | 3.36 |
| Ti | 0.01 | 0.00 | 0.04 | 0.01 | 0.00 | 0.00 | 0.02 |
| Al | 2.25 | 1.01 | 0.98 | 2.38 | 1.10 | 1.13 | 2.51 |
| Al(IV) | 0.46 | - | - | 0.57 | - | - | 0.64 |
| Al(VI) | 1.79 | - | - | 1.81 | - | - | 1.87 |
| Cr | 0.00 | 0.00 | 0.00 | 0.00 | 0.00 | 0.00 | 0.00 |
| $Fe^{2+}$ | 0.04 | 0.00 | 0.00 | 0.05 | 0.00 | 0.01 | 0.06 |
| Mn | 0.00 | 0.00 | 0.00 | 0.00 | 0.00 | 0.00 | 0.00 |
| Ni | 0.00 | 0.00 | 0.00 | 0.00 | 0.00 | 0.00 | 0.00 |
| Mg | 0.01 | 0.00 | 0.00 | 0.01 | 0.00 | 0.00 | 0.02 |
| Ca | 0.00 | 0.01 | 0.00 | 0.00 | 0.04 | 0.04 | 0.01 |
| Na | 0.12 | 0.97 | 1.02 | 0.11 | 0.90 | 0.95 | 0.24 |
| K | 0.80 | 0.00 | 0.00 | 0.86 | 0.00 | 0.00 | 0.52 |
| P | 0.00 | 0.00 | 0.00 | 0.00 | 0.00 | 0.00 | 0.00 |
| F | 0.01 | 0.00 | 0.00 | 0.00 | 0.00 | 0.01 | 0.02 |
| Cl | 0.00 | 0.00 | 0.00 | 0.00 | 0.00 | 0.00 | 0.00 |
| Total | 6.79 | 4.99 | 5.01 | 6.85 | 4.97 | 5.03 | 6.75 |

**Table 2.** Chemical analyses of white micas and plagioclases used in
the thermobarometric study of samples 1, 2 and 5.

| Unit | Lower Parautochthon (Santabaia Gp) | | | | | | | | | | | |
|---|---|---|---|---|---|---|---|---|---|---|---|---|
| Location | Bande-Celanova Dome | | | | | | | | | | | |
| Sample | 3 (Grt-Sill schist. Lower section of the Arnoia Detachment) | | | | | | 4 (Grt-Sill schist. Middle section of the Arnoia Detachment) | | | | | |
| Analysis | Bt265 | Ms267 | Gt228c | Gt214r | Pl270 | Ilm261 | Bt119 | Ms120 | Grt116c | Grt118r | Pl125 | Ilm130 |
| $SiO_2$ | 34.07 | 46.00 | 37.25 | 38.05 | 63.47 | 0.13 | 34.93 | 45.16 | 37.09 | 37.32 | 58.65 | 1.59 |
| $TiO_2$ | 1.45 | 0.55 | 0.23 | 0.05 | 0.04 | 55.73 | 2.22 | 0.60 | 0.03 | 0.00 | 0.05 | 52.05 |
| $Al_2O_3$ | 19.64 | 37.49 | 21.37 | 22.01 | 22.49 | 0.00 | 20.28 | 35.41 | 21.57 | 21.71 | 25.22 | 0.67 |
| $Cr_2O_3$ | 0.01 | 0.05 | 0.00 | 0.03 | 0.00 | 0.01 | 0.04 | 0.03 | 0.00 | 0.00 | 0.00 | 0.09 |
| $Fe_2O_3$ | 0.10 | 0.00 | 0.00 | 0.00 | 0.00 | 1.73 | 0.00 | 0.00 | 0.00 | 0.00 | 0.00 | 0.00 |
| FeO | 22.67 | 0.65 | 22.86 | 32.11 | 0.03 | 39.68 | 18.02 | 0.79 | 30.83 | 30.15 | 0.17 | 39.30 |
| MnO | 0.64 | 0.03 | 12.72 | 5.00 | 0.00 | 2.76 | 0.15 | 0.00 | 4.54 | 5.61 | 0.00 | 2.59 |
| MgO | 7.68 | 0.38 | 1.17 | 2.18 | 0.00 | 0.00 | 8.59 | 0.59 | 2.67 | 2.10 | 0.03 | 0.00 |
| CaO | 0.09 | 0.03 | 4.33 | 2.02 | 4.20 | 0.08 | 0.02 | 0.02 | 1.43 | 1.06 | 5.30 | 0.16 |
| $Na_2O$ | 0.04 | 1.33 | 0.00 | 0.03 | 9.39 | 0.05 | 0.16 | 1.04 | 0.04 | 0.01 | 8.39 | 0.00 |
| $K_2O$ | 8.14 | 9.33 | 0.00 | 0.00 | 0.10 | 0.02 | 9.20 | 9.58 | 0.00 | 0.00 | 0.11 | 0.38 |
| Total | 94.53 | 95.84 | 99.93 | 101.48 | 99.72 | 98.46 | 95.34 | 93.22 | 98.20 | 97.96 | 97.92 | 97.17 |

**Table 3.** Chemical analyses of mineral phases used in the thermobarometric study of
samples 3 (3c, 3r, Grt core and rim compositions, respectively) and 4 (4c, 4r, Grt core and
rim compositions, respectively).



| Unit | Mid-crustal section of the CIZ Autochthon | | | | | |
| Location | Villadepera Antiform | | | | | |
| Sample | 6 (Grt-Sill schist) | | | | | |
| Analysis | Bt249 | Ms248 | Grt188c | Grt172r | Pl257 | Ilm245 |
| SiO$_2$ | 36.12 | 46.65 | 36.47 | 37.41 | 64.59 | 0.05 |
| TiO$_2$ | 1.43 | 0.34 | 0.11 | 0.09 | 0.04 | 54.28 |
| Al$_2$O$_3$ | 20.80 | 37.88 | 21.92 | 22.71 | 21.82 | 0.04 |
| Cr$_2$O$_3$ | 0.03 | 0.00 | 0.00 | 0.00 | 0.00 | 0.00 |
| Fe$_2$O$_3$ | 0.00 | 0.00 | 0.00 | 0.00 | 0.00 | 0.00 |
| FeO | 18.92 | 1.21 | 27.57 | 29.33 | 0.35 | 44.40 |
| MnO | 0.19 | 0.03 | 5.83 | 0.91 | 0.05 | 0.77 |
| MgO | 9.30 | 0.47 | 1.16 | 1.16 | 0.01 | 0.23 |
| CaO | 0.03 | 0.07 | 6.71 | 8.29 | 2.72 | 0.11 |
| Na$_2$O | 0.21 | 0.99 | 0.00 | 0.02 | 9.49 | 0.01 |
| K$_2$O | 9.52 | 8.92 | 0.00 | 0.00 | 0.84 | 0.00 |
| Total | 96.55 | 96.56 | 99.77 | 99.92 | 99.91 | 99.89 |

**Table 4.** Chemical analyses of mineral phases used in the thermobarometric study of sample 6 (6c, 6r, Grt core and rim compositions, respectively).