# Peer review of "From subduction to collision in the Parautochthon and autochthon of the NW Variscan Iberian Massif"

_Solid Earth, 2020_

## Referee Comment (RC1) · Anonymous Referee #1 · 28 Mar 2020

This manuscript presents new data and interpretation of the Parautochthon sections in NW Spain. In particular, the existence of HP-LT relics led the author to suggest that the uppermost parautochthonous nappe is indeed another nappe of the Lower Allochthon. Moreover some PT conditions are presented both for the Parautochthon and the Autochthon and new tectonic contacts are proposed. The work is worth published in this special paper after some important issues being solved. I have provided a commented PDF, with details and doubts.

At a first view the idea is appealing, but it is not surprising that some parts of the Lower Allochthon appear imbricated around the uppermost section of the parautochthonous

pile. As cited by the authors, several contractional and extensional pulses overlap in the Iberian Massif and reactivation is somehow frequent. In this context the authors present three small maps without any single detail to explain the reinterpretation of previous and mixing different criteria to define new Lower Allochthonous slices. For example in the Figure 1 the Bragança complex Lower allochthon is enlarged by including part of Parano Gp at the top of the Verin synform, but in the Ordenes and Cabo Ortegal complexes this new Lower allochthon is classified out of the Allochthon with a confusing name: Uppermost Parautochthon/Lowermost Allochthon nappes. In the first case the presence of alkaline/peralkaline gneiss appear to be the criteria to include those rocks in the allochthonous ensemble. Meanwhile in the case of the red unit right below the Lalin-Forcarei thrust is the presence of albite porphyroclast with rare white mica inclusion. The Figure 2 provides some zoom into the LFT area, but not real structural details beyond the unfortunate location of samples 1 and 2 on a NE-SW fault to the E of the Forcarei synform. Late offset have been mapped in the past in this area in connection with those faults: is it possible your samples (1&2) to be part of the Forcarei unit as the result of the offset of one of those faults?

In addition a strong sinistral strike-slip shear zone is widely visible in the western limb of the Forcarei synform as well as in the vicinity of the Beariz granite (Gonzalez Cuadra et al, 2006; Fernandez et al. 2011). Why are those data not incorporated into the discussion and interpretation? On the other hand the authors show in the Fig.2 two different foliations and stretching lineations related to C1 and E1 stages. What criteria have been used to distinguish between them? There is neither description nor microstructural analysis to confirm it. Besides, it is clear that E1 lineations in the LFT area are parallel to the Carrio recumbent fold, so how do you know that those lineations are stretching/transport directions and not intersection lineations? Microstructural analysis of Fernandez et al 2011 points to a composite fabric (i.e. intersection lineations), so please explain those points.

Similarly the definition of new tectonic boundaries like the Arnoia detachment and the

[Figure]

Fumaces thrust although appealing they are not supported by the information provided by the authors. It is critical to show some detailed map with structural data and showing, for example, the telescoped isograds. It is a very good contribution but needs to be supported by data. Similar aspects can be objected to in the case of the Fumaces thrust or the new interpretation of the LFT as an extensional detachment (data?). What is the basal thrust of the Bragança complex?

Overall the findings of the authors are important but need to be explained with more data, and discuss in depth previous contributions. A special point is the definition of the Parauthochton sequence; it is confusing and need some reorganization, including a critical presentation of the differences Spanish and Portuguese parauthochthon (there's some up-to-date papers included in the Quesada&Oliveira 2019 book, for example).

Is particularly worrying an ill-advised use of argument in several parts of the manuscript than must be corrected before to considering it for publishing. For example in the introductory part of the manuscript we can find several circular arguments (lines 78, 100), where data that are part of the results and discussion of the manuscript are included as part of the introduction. Please, do no mix up introduction (previous, published, consolidated scientific knowledge) with Results (new data presented in this paper) and Discussion (interpretation of the Results confronted or not to the scientific mainstream). The problem persists along the discussion where a simple presentation of a new contact is used as a demonstration of its existence (see Line 238). The abrupt end of the manuscript in a sort of condensed discussion-conclusion chapter does not help to clarify the doubts.

I encourage the authors to review the manuscript and introduce solid arguments and scientific data to fully support their ideas, on the other hand very interesting, with the inclusion of more detail and a deeper discussion of the previous literature avoiding the use of circular reasoning.

[Figure]

Please also note the supplement to this comment:
https://www.solid-earth-discuss.net/se-2020-25/se-2020-25-RC1-supplement.pdf

---

## Author Comment (AC1) · 7 Apr 2020

Reply to the comments by Anonymous Referee #1 (28 March 2020)

We want to thank the work and constructive comments made by Anonymous Referee #1. We will try to reply here the general comments. Most of the suggested corrections in the RC #1 annotated PDF supplement are reasonable and will be accepted in a corrected version of the manuscript.

1) On the comment "without any single detail to explain the reinterpretation of previous [tectonic contacts] and mixing different criteria to define new Lower Allochthonous

slices":

We probably have not been clear enough in the text. Following point by point on the proposed examples:

a) About "...the Bragança complex Lower allochthon is enlarged by including part of Paraño Gp at the top of the Verín synform [...] the presence of alkaline/peralkaline gneiss appear to be the criteria to include those rocks in the allochthonous ensemble":

We tried to explain in the text these characteristics as suitable criteria for a correlation that we consider indubitable:

- Cartographic continuation between the top of Verín and the lower structural levels (Basal Units for some authors) of the Centro-Trasmontan Domain (allochthon), including field evidence of continuity between the Centro-Trasmontan Thrust and the Fumaces Thrust. We report the presence of HP-LT rocks in the Basal Units of the Centro-Trasmontan Domain.

- Presence of albite-porphyroblast schists in the Verín Synform, which are similar to the most abundant (volumetrically) lithology that can be found in the Basal Units (HP-LT) of the NW allochthonous massifs, and mostly lacking in the rest of the allochthons.

- Presence of alkaline gneiss in the top of the Verín Synform, including not previously described riebeckite-bearing types, which are also exclusive for the Basal Units of the NW allochthonous massifs.

b) About "Meanwhile in the case of the red unit right below the Lalin-Forcarei thrust [the criterion] is the presence of albite porphyroclast with rare white mica inclusion":

Referee #1 is right: the presence of albite-porphyroblast (not -clast) schists and the calculated PT conditions on them are here the single element to propose a possible correlative to the Basal Units of the allochthonous complexes.

c) The "confusing name: Uppermost Parautochthon/Lowermost Allochthon nappes"

is deliberately employed just because both interpretations are possible depending on the preferred criteria to apply: parautochthonous lithostratigraphic similarities versus allochthonous tectono-metamorphic features. One example of this duality is what we have proposed for the Ramallal Phyllonites and some related rocks beneath Cabo Ortegal Complex, which are considered allochthonous by some authors or parautochthonous by others. Another good example is the lower structural levels of the Centro-Transmontan Domain (sensu Ribeiro, 1974), whose possible parautochthonous nature has been recently proposed (Dias da Silva et al., 2014a, b, 2016).

2) About the comment "The Figure 2 provides some zoom into the LFT area, but not real structural details beyond the unfortunate location of samples 1 and 2 on a NE-SW fault to the E of the Forcarei synform. Late offset have been mapped in the past in this area in connection with those faults: is it possible your samples (1&2) to be part of the Forcarei unit as the result of the offset of one of those faults?"

You are right about it. The work does not include detailed structural descriptions and data beyond localized stretching lineations with criteria of relative movement. It was out of our scope. In relation to the possibility of a NE-SW late fault repeating Forcarei materials, those levels of albite-porphyroblast schists have been always interpreted to belong to the Paraño Group. Moreover, you may consider that your proposal also would need some additional tectonic elements, such as some part of a klippe plus a N-S trending synform or a N-S trending fault that are not recognized.

3) With regard to the comment "In addition a strong sinistral strike-slip shear zone is widely visible in the western limb of the Forcarei synform as well as in the vicinity of the Beariz granite (Gonzalez Cuadra et al, 2006; Fernandez et al. 2011). Why are those data not incorporated into the discussion and interpretation?"

Yes, you are right. We have no data on kinematics west of Forcarei and even paid too much little attention to those works. We will correct it, thank you. About the content of your comment, Fernández et al., 2011 propose the development of a flat-lying S2

foliation during D2 tangential tectonics, where stretching lineation L2 indicates early top-to-the-south thrusting preserved in its eastern limb (just as González Cuadra et al., 2006), but late top-to-the-north shear in its western limb. Thus, both limbs show sinistral criteria once uprighted by D3 folding, and our LE1 observations from the eastern limb of Forcarei could be equivalent to their late D2 stage ones. An early D2 top-to-the-south thrusting would correspond to our C2, but then, the Lalín-Forcarei Thrust which is top-to-the-E in Lalín (Martínez Catalán et al., 1996; 2002), would be (reactivated?) top-to-the-S in Forcarei. Another possibility is that top-to-the-S and top- to-the-N could be conjugate movements of our E1 stage, as proposed for the Redondela-Beariz Detachment in Díez Fernández et al. (2012b), and for Bande-Celanova Dome in this manuscript. This one is a more likely option, as top-to-the-south (sinistral) shearing in the eastern limb affects the Beariz granite as well (González Cuadra et al., 2006). Top-to-the-north kinematic criteria are also found by Gloaguen et al., 2014 (and references therein) to the east of Forcarei, where the authors relate both, stretching lineations (our E1) and upright folds (our C3), with a post-D2 (main thrusting) stage D3.

4) "On the other hand the authors show in the Fig.2 two different foliations and stretching lineations related to C1 and E1 stages. What criteria have been used to distinguish between them? There is neither description nor microstructural analysis to confirm it."

Yes, you are right. C1 criteria for the Lalín-Forcarei Thrust are taken from Martínez Catalán et al., 1996; 2002 (quartz c-axis, asymmetrical boudins and S-C microstructures), and point to a top-to-the-east sense of thrusting for the Lalín-Forcarei Basal Unit in the Lalín Synform, related to early stages of collisional thickening. It is commonly accepted and we do agree with it. You are right and we must specify this in the figure caption, the text or both. However, our new observations in the "red unit" beneath Lalín point to a top-to-the-north shear sense, just similar to our data in eastern Forcarei and other areas. Our field criteria are based on asymmetrical biotite mica-fish, boudinaged quartz veins and drag-folds. We agree that this information was skipped from the text and we will correct it. Respecting to the discrimination between both

sets of orientations, we have tried to make it clear in the manuscript that mainly top-to-the- north E1 deformation shows a subtractive character where there are expected stratigraphic markers that are (tectonically) missing (Bande-Celanova Dome) and it is sometimes (when deep enough) related to low-P, high-T metamorphic overprinting (Arnoia Detachment), so it very likely represents syn-collisional extension, as opposed to C1 thickening.

5) "Besides, it is clear that E1 lineations in the LFT area are parallel to the Carrio recumbent fold, so how do you know that those lineations are stretching/transport directions and not intersection lineations? Microstructural analysis of Fernandez et al 2011 points to a composite fabric (i.e. intersection lineations), so please explain those points."

Well, that is not exactly that way. Fernández et al., 2011 points to two different lineations: an intersection lineation L3 related to their D3 upright folding (our C3), and the previous L2 stretching lineation that we have commented above. Both lineations usually show the same trend and are sometimes difficult to discriminate in the internal zones of the Iberian Massif, especially in sections affected by late, tight, upright folds such as Forcarei, where axial-plane schistosity may be very penetrative. Not the case in the Lalín Synform, which is relatively open. Anyway, the Carrio recumbent fold is above the LFT, that flattens and crosscuts its basal reverse limb, but our E1 lineations have been observed in Lalín from the LFT downwards. The asymmetrical shapes of deformed elements, showing their best asymmetry over sections parallel to the lineations, would also indicate that are not intersection lineations. Similar stretching lineations, which occur alongside top-to-the-north shear sense criteria, are common in other areas of the GTMZ (Marcos and Farias, 1999; Martínez Catalán et al., 2002; Gómez Barreiro et al., 2010; Fernández et al., 2011; Díez Fernández et al., 2012b; Gloaguen et al., 2014; this work).

6) "Similarly the definition of new tectonic boundaries like the Arnoia detachment and the Fumaces thrust although appealing they are not supported by the information provided by the authors. It is critical to show some detailed map with structural data and showing, for example, the telescoped isograds. It is a very good contribution but needs to be supported by data. Similar aspects can be objected to in the case of the Fumaces thrust or the new interpretation of the LFT as an extensional detachment (data?)."

You are probably right. It means at least two new figures (Arnoia, Fumaces...) that we will try to include. Anyway, a previous sketch with the telescoped isograds (excluding kyanite) can be found in Barrera Morate et al. (1989). We will work on it, thanks. Respect to the LFT, we do not propose that it is an extensional detachment as you say, but there are kinematic criteria indicating a possible extensional top-to-the-north reactivation of the top-to-the-E thrust (see Fig. 2). The complex nature of the structure is also pointed out by the apparent sinistral indicators that you cite above about Lalín (and our observations). From your comment, we realize that our treatment of LFT in Fig. 2 is unfortunate, so we will change it. Thanks again.

7) "What is the basal thrust of the Bragança complex?"

We think this point is clear in the manuscript: the basal thrust of the Centro-Trasmontan Domain by Ribeiro (1974) (wonderful map), whose possible continuation into Verín we propose to be the Fumaces Thrust. Our work shows top-to-the-ENE criteria, probably corresponding to the same stage as the LFT.

8) "Overall the findings of the authors are important but need to be explained with more data, and discuss in depth previous contributions."

We will try to add detailed maps of the Fumaces Thrust and the Arnoia Detachment. We will include the references that you cited above and some discussion about them in the terms of this reply.

9) "A special point is the definition of the Parauthochton sequence; it is confusing and need some reorganization, including a critical presentation of the differences Spanish and Portuguese parauthochthon (there's some up-to-date papers included in the

Quesada&Oliveira 2019 book, for example)."

We are afraid that the definition of the Parautochthon sequence, as you ask for, is not simple. It is explained in the introduction that there are several different descriptions from geologically (and other reasons) disconnected sections. Correlation between them will be for long a matter of debate. We think that the two most representative and accepted subdivisions in Spain and Portugal (you can search both countries geological survey webs) are correctly exposed in the text, though it can be hard to follow for many readers. Even correlation just between this two is problematic. About the Quesada and Oliveira 2019 book, the corresponding chapter is a great review, but the number of up- to-date (e.g. post-2014) regional references cited is less than ours. Anyway we will check it, thank you.

10) "Is particularly worrying an ill-advised use of argument in several parts of the manuscript than must be corrected before to considering it for publishing. For example in the introductory part of the manuscript we can find several circular arguments (lines 78, 100), where data that are part of the results and discussion of the manuscript are included as part of the introduction. Please, do no mix up introduction (previous, published, consolidated scientific knowledge) with Results (new data presented in this paper) and Discussion (interpretation of the Results confronted or not to the scientific mainstream). The problem persists along the discussion where a simple presentation of a new contact is used as a demonstration of its existence (see Line 238). The abrupt end of the manuscript in a sort of condensed discussion-conclusion chapter does not help to clarify the doubts."

You are probably right about taking out from the introduction both references in lines 78-79 and 100-102, and lines 238-242 from the tectono-metamorphic constraints. Thank you. About discussion & conclusions chapters, it is not a rare practice, but we will consider separating them for the final version as you suggest.

11) "I encourage the authors to review the manuscript and introduce solid arguments

and scientific data to fully support their ideas, on the other hand very interesting, with the inclusion of more detail and a deeper discussion of the previous literature avoiding the use of circular reasoning."

We agree on the interest of presenting more detailed maps, particularly from both new structures. Probably we will replace the equivalent old figures in the final version. We acknowledge very sincerely your suggestions. About more detail in the discussion, please note the regional amplitude of the work and the multidisciplinary matters to deal with from more than 70 regional references currently used here. Perhaps your idea is more appropriate for a larger format.

References not included in the manuscript

Fernández, F. J., Díaz-García, F., and Marquínez, J.: Kinematics of the Forcarei Synform (NW Iberian Variscan belt). Geol. Soc. London, Spec. Publ., 349, 185–201, doi: 10.1144/SP349.10, 2011.

Gloaguen, E., Branquet, Y., Chauvet, A., Bouchot, V., Barbanson, L., and Vigneresse, J. L.: Tracing the magmatic/hydrothermal transition in regional low-strain zones: The role of magma dynamics in strain localization at pluton roof, implications for intrusion-related gold deposits. J. Struct. Geol., 58, 108–121, doi: 10.1016/j.jsg.2013.11.006, 2014.

González-Cuadra, P., Díaz García, F., and Cuesta Fernández, A.: Estructura del granito de Beariz (Ourense, Galicia). Geogaceta, 40, 151–154, 2006.

---

## Referee Comment (RC2) · Alicia López-Carmona (Referee) · 14 Apr 2020

The manuscript by Rubio Pascual et al presents a thermobarometric study on several samples from the so called Autochthonous and Parautochthonous domains (NW Iberian Massif) concluding that the uppermost parautochthonous nappe, composed of Upper Paleozoic syn-orogenic rocks with high-pressure relicts, is another nappe of the so called Lower Allochthon. Additionally, several tectonic schemes are included.

The manuscript is well written, the length of the sections is balanced, and the figures and tables are well presented (see specific comments on the figures contents). However, the manuscript is not well organized, the content of some sections (2, 3 and 4)

[Figure]

does not correspond to the headings and it is difficult to distinguish what are the (mainly structural) new contributions of this work. Besides, important regional references are missing. The metamorphic study is relevant (see comments and specific observations below) but it is not fully supported.

Specific comments by sections:

Section 1;

L43; "...and other correlatable Variscan suture outcrops of the Iberian Massif...". The OMZ has been classically considered a composite terrane with domains of different origin, separated by strike-slip shear zones, but without a proven tectonic relation between them (see e.g. Simancas et al., 2016; Azor et al., 2019). Recently, this view has been re-interpreted (Díez Fernández and Arenas, 2015; Arenas et al., 2016; Díez Fernández et al., 2016; Abati et al., 2018) suggesting a common origin for the OMZ and the Allochthonous Complexes of NW Iberia, and a correlation between their different units has been proposed. Currently, these correlations are highly debated within the Iberian geological community. Please, reformulate these lines and include the references of the authors who have presented arguments that do not favour these correlations.

L47-48: "...some works (Pitra et al., 2010...) and new data from the NW of the Iberian Massif, point to a more complex scenario in which part of the considered parautochthonous section experienced HP-LT conditions before the onset of Barrovian metamorphism". Albite-bearing micaschists (from the Mauves Unit) were ascribed to the Parautochthon by Ballèvre et al. (2009) and Pitra et al. (2010), and subsequently attributed to the Lower Allochton by Ballèvre et al. (2014) because of its similarity to the Spanish outcrops. These authors do not ascribe Barrovian metamorphism to the Mauves Unit. They just used a different nomenclature, and the Variscan units in the Iberian and Armorican Massifs were later unified (in Ballèvre et al., 2014). Please interpret correctly and cite the appropriate references.

Section 2.1.1 includes a good synthesis on the metamorphism of the different units of

the Allochthonous Complexes. In section 2.1.2 the same synthesis is expanded with the description of the domain structure, together with the ages of specific structures, and regional deformation phases are presented. Section 2.2 focuses on the depiction of the regional deformation phases and the metamorphism is discreetly mentioned. I think it would bring accuracy to the manuscript to sort these descriptions and follow the same scheme in all sections and subsections. Furthermore, if this description is accompanied by a table/figure that summarises the relationship between the foliations (which are introduced in section 4, along with all the other nomenclature), the deformation phases, the metamorphic events and the (relative/absolute) ages, it would be of great help for potential readers. The nomenclature used by the authors was previously proposed by Martínez Catalán et al. (2014) and refined by Dias da Silva et al. (2017, 2020) and Azor et al. (2019). Please, include the missing references.

In section 3 it seems that the structural arguments that support the proposal of a new tectonostratigraphic sequence are presented. What are the original contributions regarding field and structural data, beyond the simplified schemes presented in figures 2, 3 and 5? Are some taken from references? Please clarify this aspect in the text and include in figures 2, 3 and 5 (or in a new one) cross sections, structural data and images of the outcrops that support this interpretation (e.g. L160, L182…). It is tedious to follow the reasoning with so much regional terminology. Given the proposed conclusion, it is convenient to standardise the internal divisions of the Parautochthon and to support the arguments presented by establishing their connections with all the existing tectonostratigraphic models. L 150-152 if this interpretation refers to that of Días da Silva et al. the interpretation mentioned in this text has nothing to do with the one deduced from these authors. According to Días da Silva & Clavijo (2010): "Under the main Trás-os-Montes thrust plane, in the easternmost region of the Morais Allochthonous Complex, a geologic unit has been identified. It shows syn-tectonic S2-related andalusite blastesis, representative of low-pressure thermal metamorphism. In the studied sector this metamorphism affects essentially the black slaty lithologies present in Neoproterozoic to Silurian formations." Then, Días da Silva et al. (2014a, b

and 2016) present a new definition of the tectonostratigraphy and structural boundaries of the Parautochthon in the eastern rim of the Morais Allochthonous Complex (NE Portugal) based on a comprehensive characterization of the Saldanha and Mora volcanic complexes and no mention to HP-LT rocks is made.

Section 4 is entitled "... constraints" but what is presented is a synthesis of the background classified by types of metamorphism describing foliations, metamorphic events, and deformation phases. In this synthesis, what appears to be original contributions of this work are disguised, going unnoticed. Like the great contribution described in section 4.1, which is the report of the ky-zone in this area for the first time. L226: "For several reasons, the HP nappe of the SACG could have a likely continuation in the uppermost part of the Paraño Group in the Verín Synform". For what reasons? Please, specify.

Section 5; Since one of the contributions of this work is the description of the tectonothermal evolution of the studied samples, and the P-T quantification by means of conventional thermobarometry, please, relate the petrographic images in figure 4, and the minerals chemical analyses in table 2 (with backscattered images if necessary due to resolution restrictions) and provide a report of the mineral chemistry that matches the P-T constraints. Also, in figure 4, indicate to which sample does each image corresponds. L265; I do not doubt that the authors' conclusion is correct, but I think that it is essential to develop it further and it must be supported by more robust arguments. For example, when the authors refer to "show high silica contents (Table 2) and a mineral chemistry compatible with HP-LT metamorphic conditions" it is important to develop that mineral chemistry. Si values in muscovite are compatible with those reported in HP-LT rocks. Please show images of the albite porphyroblasts inclusions showing white mica crystals. Describe the albite content in plagioclase and compare this content with those in the samples from the Lower Parautochthon. For example, in the Ceán Schists, albite porphyroblasts, which contain white mica inclusions with similar Si values in muscovite, were thought to contain a relict foliation S1 of HP-LT (López-

Carmona et al., 2010). However, after studying numerous samples and analysing the included garnets, it was discovered that this foliation is an M2 (López-Carmona et al., 2013), still under HP conditions. In contrast, in the Cambre amphibolites (texturally) similar porphyroblasts contain an S3 (M3). How are your albite porphyroblasts related to those of these rocks belonging to the Basal Units/Lower Allochthon (for example, in the Santiago and Ceán Units?)

Minor comments:

Please unify abbreviations consistently in the text (see text underlined in green in the annotated version of the manuscript) Please, unify the spelling in the terminology referring to " parautochthon". It is written in different ways through the manuscript, with and without capital letters.

[Figure]

**Supplement:**

[revised manuscript text omitted]

---

## Author Comment (AC2) · 20 Apr 2020

Reply to the comments by Referee #2, Alicia López Carmona (14 April 2020).

We sincerely acknowledge the review work and constructive comments made by López Carmona. We will try to reply here the general comments. Really most of the suggested corrections in the RC #2 annotated PDF supplement are correct and reasonable, specially those referred to clearly distinguishing new contributions from previous works by other authors, and including formal and English corrections that we truly appreciate and will be introduced in the corrected version of the manuscript.

[Figure]

About your introductory resume:

In your lines 3-4 you write "...concluding [the authors] that the uppermost pa-rautochthonous nappe, composed of Upper Paleozoic syn-orogenic rocks with high-pressure relicts..." but we do not conclude, even mentioned, that in any place: In our lines 2-23 we write that the uppermost nappe is made of Lower Paleozoic rocks, and only the other upper and the lower parautochthonous nappes are formed by both, Lower Paleozoic pre-orogenic and Upper Paleozoic syn-orogenic rocks. It is the same in L86-90. Besides, in L112-114 we write about the syn-orogenic deposits covering the CIZ autochthon.

On the specific comments by sections:

Section 1

L43 about correlation between the GTMZ and the OMZ:

You have to note that we say "other correlatable Variscan suture outcrops", avoiding using "same suture". The cited GTMZ and OMZ outcrops are, at least, correlatable in matter of HP conditions and age of HP metamorphism. However, in view of your comment and other possible misunderstandings by other readers, we will better eliminate the whole reference to other suture outcrops. Thank you.

L47-48 about misinterpretation of references from the Armorican Massif:

We never wrote that Pitra et al. (2010) (or Ballèvre et al.) ascribed Barrovian metamor-phism to the parautochthonous Mauves Unit. Effectively, you are right, it was consid-ered as a parautochthonous unit beneath the Lower Allochthon in Ballèvre et al. (2009) and Pitra et al. (2010) and it was later attributed to the Lower Allochthon in Ballèvre et al. (2014). Pitra et al. (2010) obtain wide metamorphic conditions < 10 kbar and < 540 °C for the matrix foliation of the albite-bearing schists, but they textually suggest an earlier higher pressure event related to the inclusions in the albite porphyroblasts, and that is our reason to cite the work. Effectively, there is a change of nomenclature

in Ballèvre et al. (2014), and both parautochthonous and Basal Units are then grouped as Lower Allochthon, but the question is not that, because the albite-bearing schists of Mauves, former parautochthon, are anyway correlated by you (as co-author) with the HP Basal Units of Galicia. So that, as in the case of our work, there are units sometime considered as parautochthon that experienced HP-LT metamorphism, and it was not always considered that way in some places such as the GTMZ, even my self as co-author, for example, in a reference cited in the manuscript. We will introduce the references by Ballèvre et al. (2009; 2014), thank you.

Section 2

About sorting of descriptions following the same scheme in all sections and subsections:

Yes, you are right, thank you. We will try to homogenize the content of the subsections, at least for the Parautochthonous Domain and the autochthon. The allochthons are a very more complex issue, and only the Basal Units were of interest in our work.

About including a table summarizing deformation phases, foliations, metamorphic events and ages:

It is a very good idea, thank you. We will include it.

About missing references:

You are right, references will be included. Thank you very much.

Section 3

About clarifying original contributions or taken from reference structural data in the text, figures 2, 3 and 5 (or in a new one), cross sections, structural data and outcrop images:

You are right. We must clarify what is what and to include the references when necessary. We are going to include new figures with more detailed structural map at least for the Fumaces Thrust and the Arnoia Detachment areas. Including cross sections

in these new figures would be a good idea and we will try to do it. We consider that adding a new table in section 2 and at least two new figures in section 3 are necessary enough. Including also some outcrop images probably does not deserve the length increase, but as you also ask for more images further in Section 5, we will try to include some of them, thank you.

About "tedious" regional terminology, standardising the internal divisions of the Parautochthon and establishing their connections with all the existing tectonostratigraphic models:

There are few tectonostratigraphic models apart from those derived from Rodrigues et al. (2003) in Trás-os-Montes, which is already treated in the text. Most of the Parautochthon in Galicia is yet commonly considered a continuous stratigraphic sequence (with little exceptions: Valverde Vaquero et al. (2005) in Cabo Ortegal; Farias and Marcos, 2019 in the Alcañices Synform). We have tried to avoid as much as possible to name or describe formations. Our study areas in the Parautochthon are focused in the SACG, Verín Synform, Bande-Celanova Dome and, with a minor extent, the Cabo Ortegal area. The three first zones have been commonly described in literature following the Santabaia-Nogueira-Paraño Gps stratigraphic classification (there was not any structural subdivision), and it has been treated in our work. The fourth one has an old former different stratigraphic subdivision (Loiba and Queiroga series, Arce Duarte and Fernández Tomás, 1976; Arce Duarte et al., 1977) that we have not cited expressly, but we used more recent works (originally based on that subdivision, references therein) that present chronological data with structural implications which are our real interest and so are exposed in our manuscript. Other local stratigraphic or petro-stratigraphic units have been cited exclusively because of its structural or metamorphic relevance. To introduce a comprehensive table of formations is out of our scope, and it inevitably would lead to an even more "tedious" list of local names.

About L150-152 and misinterpretation of Dias da Silva et al. (2014a, 2014b, 2016):

[Figure]

Please, contemplate carefully this question. The limit between the Centro-Trasmontan Domain (allochthons) and the Peri-Trasmontan Domain (the Parautochton) was first established in the Main Trás-os-Montes Thrust of Ribeiro et al. (1990), a well mapped structure in the whole region since Ribeiro (1974). The works by Dias da Silva et al. in the eastern area of Morais concluded that most of the previously considered Lower Allochthon (e.g. Ribeiro et al., 2006) is currently a part of the parautochthonous section. The lawsonite-bearing rocks (HP) described in that area by Schermerhörn and Kotsch (1984) -and cited in their work-, were then interpreted as belonging to a possible, thin, uppermost unit of the top of the Parautochthon (please see Fig. 1 in Dias da Silva et al., 2014b). This conclusion was also extended to the rest of Morais (except for the Valbom dos Figos blueschists and Mascarenhas rocks), and it was also extended to the Bragança Complex, where all the former Lower Allochthon was considered a part of the Parautochthon in Dias da Silva et al. (2014a, please see Fig. 1), or alternatively it was restricted in 2014b (please see Fig. 1) to the uppermost part of the previously commonly considered LA section, with not any tectonic boundary described there. So that, we consider to be right when we write that, according to those works, i) most of the former Lower Allochthon is considered as a part of the Upper Parautochthon, ii) there is not any other recognized basal thrust structure for the Lower Allochthon of the Bragança Complex, and iii) some of the HP-LT rocks of the Morais Complex stay in the Upper Parautochthon. This said, it must be clear that the main results of Dias da Silva et al. in eastern Morais are not in contradiction with ours, as our conclusion is also open to consider an uppermost parautochthonous unit with HP rocks.

Section 4

About the title "constraints" when it is a "synthesis of the background" including original contributions disguised/unnoticed:

Yes, you are right. Thank you very much. We will change the title to "Tectonometa-morphic background and new data" or similar, and then we will separate clearly both

blocks of information in every subsection.

About L226, to specify the reasons for a likely continuation of the HP nappe of the SACG in the uppermost part of the Paraño Group in the Verín Synform:

Well, the next sentences in the manuscript pretend to explain those reasons. L227 "Farias (1990) also mentions there the presence of albite porphyroblasts with an internal schistosity of aligned micro-inclusions of quartz, chlorite and white mica", that is, the presence in both places of albite porphyroblast-bearing schists. L228-229 "The Fumaces Thrust of Verín has outcropping continuity into the basal thrust of the lower allochthonous Centro-Trasmontan Complex with HP-LT rocks", and L230-231 just add another element of possible continuity between the uppermost part of the Paraño Group in the Verín Synform and the LA in Portugal. From this, the upper part of the Verín Synform seems to have continuity in Portugal through an unit that have been identified for many authors in Portugal as a Lower Allochthon that includes HP-LT rocks, and both sections present alkaline-peralkaline rocks. Thus, uppermost SACG and uppermost Verín Synform-LA in Portugal seem to share not only some characteristic type of rocks, but also the early HP history. Probably you mean that both are, at last, just one reason, if you consider that the albite-bearing rocks are HP rocks, and in that case you are right, so we will suppress "For several reasons". Thank you.

Section 5

About relating the petrographic images in figure 4 and the chemical analyses in table 2, and also indicating, in figure 4, to which sample does each image correspond. To show images of the albite porphyroblasts inclusions showing white mica crystals:

Ok, if you consider it necessary, though it introduces again a problem of length. We used one single picture illustrating an albite porphyroblast, and it was chosen on textural and micro-structural more than mineralogical or probe analyses location criteria. To accomplish with your suggestion, we have to add an extra figure better than replacing pictures. We could also include then some outcrop images (that you asked for above)

Interactive
comment

to complete the figure.

About providing a report of the mineral chemistry that matches the P-T constraints. Si values in muscovite:

Maximum Si p.f.u. contents, those used in thermobarometry, are effectively shown in Table 2 as you say, with the complete oxides weight and cations. Albite porphyroblasts are small, the relic schistosity is very thin and rich in organic matter (Fig. 4 caption), garnet and even apparently epidote-free, and everything points to a really low temperature of crystallization. White mica inclusions are scarce and very few times they are sized into the beam diameter. Failed analyses yielded aberrant compositions with high Na20 content and were easy to discard.

About describing the albite content in plagioclase and comparing this content with those in the samples from the Lower Parautochthon:

The chemical compositions of porphyroblastic albite used in the thermobarometry of the uppermost nappe are also shown in Table 2. We will include in the text some synthetic information about the molecular content in all the analyses we performed: Ab% 98.44 - 99.63 in porphyroblast cores with internal schistosity, 89.94 – 95.94% in inclusion-free rims. Plagioclase is practically absent of the other pelitic rocks in the Upper Parautochthon, and it is present again only in the Lower Parautochthon rocks of medium or high metamorphic grade. In the Arnoia River section these are granoblastic to small porphyroblastic crystalls oriented in the matrix foliation (SE1). Oxides composition of the chemical analyses used in thermobarometry are included in Table 3. We can also include in the text that the Ab content in all the performed analyses is 87.06 – 93.96%.

About how are our albite porphyroblasts related to those of rocks belonging to the Basal Units:

Please note L222-224 "Albite-bearing schists are characteristic rocks in the al-
lochthonous Basal Units, where the porphyroblasts present a more advanced textural development and include more complete HP-LT mineral parageneses (Arenas et al., 1995)." We can also add that those from the SACG are of much lower size, typically below 0.5 mm. About their Ab content, we will include that it is slightly higher than the core compositions of porphyroblasts from the Santiago Unit (> 97%) and close to those from the Ceán Schists. Of course, we will include your references, thank you.

About your Minor comments:

Yes, you are right. We apologize and we will do it, thank you very much.

References not included in the manuscript or the referee's review

Arce Duarte, J. M., and Fernández Tomás, J.: Mapa Geológico de España a escala 1:50.000, Hoja 8 (7-3), Vivero. Instituto Geológico y Minero de España, Madrid, 45 pp, 1976.

Arce Duarte, J. M., Fernández Tomás, J., and Monteserín López, V.: Mapa Geológico de España a escala 1:50.000, Hoja 7 (7-2), Cillero. Instituto Geológico y Minero de España, Madrid, 47 pp, 1977.

Ribeiro, A., Pereira, E., Ribeiro, M. L., and Castro, P.: Unidades Alóctones da Região de Morais (Trás-os-Montes oriental). In: Dias, R., Araújo, A., Terrinha, P., Kullberg J.C. (eds), Geologia de Portugal no contexto da Ibéria, Univ. Évora, 85–105, 2006.
* * *